# Self-Organization of the Retina during Eye Development, Retinal Regeneration In Vivo, and in Retinal 3D Organoids In Vitro

**DOI:** 10.3390/biomedicines10061458

**Published:** 2022-06-20

**Authors:** Eleonora N. Grigoryan

**Affiliations:** Koltzov Institute of Developmental Biology, Russian Academy of Sciences, 119334 Moscow, Russia; leonore@mail.ru; Tel.: +7-(499)-1350052

**Keywords:** retina, self-organization, cellular and molecular mechanisms, development, regeneration, histotypic reaggregates, organoids

## Abstract

Self-organization is a process that ensures histogenesis of the eye retina. This highly intricate phenomenon is not sufficiently studied due to its biological complexity and genetic heterogeneity. The review aims to summarize the existing central theories and ideas for a better understanding of retinal self-organization, as well as to address various practical problems of retinal biomedicine. The phenomenon of self-organization is discussed in the spatiotemporal context and illustrated by key findings during vertebrate retina development in vivo and retinal regeneration in amphibians in situ. Described also are histotypic 3D structures obtained from the disaggregated retinal progenitor cells of birds and retinal 3D organoids derived from the mouse and human pluripotent stem cells. The review highlights integral parts of retinal development in these conditions. On the cellular level, these include competence, differentiation, proliferation, apoptosis, cooperative movements, and migration. On the physical level, the focus is on the mechanical properties of cell- and cell layer-derived forces and on the molecular level on factors responsible for gene regulation, such as transcription factors, signaling molecules, and epigenetic changes. Finally, the self-organization phenomenon is discussed as a basis for the production of retinal organoids, a promising model for a wide range of basic scientific and medical applications.

## 1. Introduction

The major objective of developmental biology and biomedicine is to address the question as to how a complexly organized organ, which is a coordinated system programmed to perform certain functions, is assembled from simple embryonic tissues. The retina, as part of the central nervous system (CNS), provides an easily accessible model system for investigating the histogenesis of the complex neural tissue. It possesses a simplified anatomical structure and consists of well-defined cell types, which is a convenient feature facilitating retinal research. Self-organization is a process that provides the formation of the complex and regular retinal structure during development in vivo and in vitro. Self-organization underlies the development and regeneration of the retina in vivo and serves as a basis for the formation of retinal 3D organoids in vitro. Despite long-term studies on various models, the phenomenon of retinal self-organization still remains poorly understood. It cannot be fully explained by any of the existing theories of biological self-organization, including those based on the widely studied molecular–genetic and epigenetic mechanisms of the process. The retinal self-organization process involves a wide range of molecules and mechanisms regulating the expression of genes in cells. Molecules of cell–cell communication, extrinsic signaling molecules, transcription factors, mechanotransduction molecules, etc., are all within this range. The biological complexity and genetic heterogeneity of the process necessitate an integrated understanding and addressing the fundamental, but still-unresolved scientific issues. Currently, the application of new models, cutting-edge molecular genetics, cultivation technologies, and interdisciplinary approaches contribute to a deeper understanding of the process of retinal self-organization.

Consideration of the general ideas about the self-organization phenomenon, which is a basic law that works in the physical, biological, and social realms, is beyond the scope of the present review. The process of self-organization in biological systems has been studied in a large number of dedicated works addressing and investigating this phenomenon from various aspects. Moreover, even formally defining self-organization in biology is challenging, in part because this term has been used in diverse contexts and with different purposes [1,2,3]. In general terms, according to the currently existing views, the concept of biological self-organization is (1) an all-important feature of living systems; (2) regarded as a process of spontaneous pattern formation; and (3) a process manifested in emerging structures that are distinguished from their environment on distinct spatiotemporal scales and in the absence of centralized control or external drivers [2,3,4,5,6,7,8].

In this review, an attempt is made to summarize the basic information on the retina’s self-organization in vertebrates and to identify the promising areas for further study of the phenomenon. In the article, the process of retinal self-organization is considered during normal development, during regeneration, and in cultivation systems including histotypic retinal cell reaggregates and retinal 3D organoids.

Understanding the retinal self-organization process and its fine mechanisms can contribute to the further construction of the fundamentals of developmental biology, as well as address a number of biomedical issues. Among the latter are those related to congenital retinal pathology, regeneration and reconstruction of the retina in vivo after damage, prevention of retinal degeneration, and the creation of retinal 3D organoids. The study of the latter is of particular interest, since it provides a wide range of opportunities for disease modeling, drug development and screening, personalized medicine, as well as for understanding organogenesis.

## 2. Retinal Structure and Function in Adult Vertebrates

The retina’s structure has a plan common for all vertebrates including humans, with some evolutionarily fixed morphological and functional features [9,10]. The retina is a highly organized, stratified neural tissue, where different types of cells are arranged into strict localizations, maintaining interaction with other cells in the 3D retinal space. In functional terms, the retina is a sensory tissue organized into cell layers with microcircuits operating in parallel and synergistically to encode visual information. All vertebrate retinas share a fundamental plan comprising five major neuronal cell classes, with cell bodies’ distributions and connectivities arranged into stereotypic patterns. The retina consists of pigment epithelium (RPE) and the neural retina (NR), represented by interconnecting layers of specialized cells. The six major types of NR cells are neurons (rods and cones as photoreceptors, bipolar cells, horizontal cells, amacrine cells, and ganglion cells). Müller glial cells (macroglia), microglial cells (resident tissue-specific macrophages), astrocytes, and oligodendrocytes are non-neuronal, NR-integrated cells (Figure 1). The retina has three nuclear layers and two plexiform layers representing synaptic contacts between retinal neurons. The outer nuclear layer (ONL) of the retina includes light-sensitive cells (rods and cones). Their outer segments exhibit topological and functional interactions with the RPE. The inner nuclear layer (INL) includes interneurons: bipolar, amacrine, and horizontal cells. Bipolar cells are involved in the visual signal transmission from photoreceptors into ganglion cells; horizontal and amacrine cells connect all cells of the retina. Ganglion cells form a ganglion cell layer, where their long axons are involved in the optic nerve formation. The outer plexiform layer (OPL) is organized into fibers and synaptic contacts between rods/cones and bipolar cells, while the inner plexiform layer (IPL) provides a connection between bipolar and ganglion neurons, as well as a horizontal connection between amacrine and horizontal neurons. Müller glia cell bodies are localized in the INL and extend their radial processes toward the outer and inner retinal limiting membranes.

## 3. Embryonic Retina Self-Organization In Vivo

### 3.1. General Concepts

The ontogeny of the vertebrate retina has been a topic of interest to developmental biologists for many decades. The retina’s development in vertebrates begins early, from the time of the division of the anterior neural plate into domains, with the specification of the so-called eye field occurring in the middle of them. The bilateral optic vesicles, being the eye anlage, are subsequently formed in this region. The retina appears in the posterior wall of the optic cup, which is formed from the optic vesicle through its invagination. Simultaneously, the eye primordium elongates in the posterior part, and its connection to the brain grows narrower, generating the optic stalk (Figure 2) [11,12]. A lineage tracing and live imaging, carried out on zebrafish embryos, allowed a detailed analysis of cell movements, including extended evagination and rim movement [13,14,15]. Retinal progenitor cells (RPCs) in the optic cup undergo active proliferation, producing the prospective RPE and NR [11,12]. Those cells that remain in the outer layer of the optic cup constitute the RPE progenitors, whereas the inner layer is composed of the NR progenitors. Each of the RPCs is multipotent and capable of producing the full range of retinal cell types subsequently. Different clones of RPCs have different combinations of precursors of the major cell types, including Müller glial cells. It is postulated that the interactions that specify the differentiation pathway of retinal cells occur relatively late in development [16,17,18,19,20,21].

The “competence model”, based on the sequential maturation of retina cell types in a certain order and as a result of changes in the RPCs’ competencies (potencies), is considered as a fundamental basis for the self-development of the vertebrate NR [19,20,21]. The sequential manner of retinal cell differentiation and its “timing”, exhibiting certain patterns, have been identified in different vertebrate species. It is usually manifested as the generation of ganglion cells, cone photoreceptors, and horizontal and amacrine cells of the retina in the early phase, overlapping with the late-phase generation of rod photoreceptors, bipolar cells, and Müller glia [22]. To date, the main retinal time course, similar to that found in other vertebrates, has been described from the developing human retina using RNA-Seq analysis [23,24]. The competence model, reflecting the intrinsic pattern of cell diversification in the developing retina, has been further supported by the recently collected extensive information on the expression of transcription factors (TFs) and signaling molecules that jointly determine the hierarchy of RPCs [25,26,27,28,29,30,31] (see below) (Figure 3).

The laminar structure of the retina (Figure 1) is formed during cell maturation. Its formation in vivo occurs not only through the implementation of internal cell competencies and RPCs’ interactions with each other, but also through external signaling. At the earliest stages, the retina is formed largely through self-organization, a process whose mechanisms are not fully understood. There is some disagreement between the two theories existing today [32]. The first, the above-mentioned competence model, is the theory of RPCs changing competencies, which provides different cell clones for the development of certain cell types at a certain time [22] (Figure 3). This model suggests that during retinogenesis, RPCs acquire and then lose the ability to produce certain cell types in accordance with some internal timing. An assumption has been made that such a behavior of RPCs is regulated and controlled by the cells’ mutual positive/negative feedback through intercellular adhesion, communication, and physical factors, as well as signaling from other developing eye tissues [19,20].

However, in vitro studies on developing zebrafish and rat retinal tissue have shown that the “histogenesis of fates” observed in the overall population is accounted for by the variability of cell determination within clones and that cell fate variability among clones is likely to have a partially stochastic explanation [32,33,34]. To confirm this, information is provided that after the elimination of progenitor cells of one or the other cell type in the zebrafish and mouse embryonic retina in vivo, the remaining cells are able to laminate in the correct order [35,36,37]. Thus, lamination of the retina was restored in *Chx10*, *Kip1* double-null mice in the absence of bipolars [35]. In the transgenic zebrafish, an IPL-like neuropil still formed in cellularly simplified retinas consisting of only bipolars and photoreceptors. Remarkably, in this presynaptic-only neuropil, axons of bipolars could still make presynaptic structures and display sublaminar organization of their axonal terminals [36].

There are also data showing that retinal explants, taken at different time points of development and, therefore, containing different populations of RPCs, reproduce in vitro the same size and composition of clones as in vivo [33]. These facts are evidence that both the competence of RPCs and their fate are stochastic to a certain extent and, at the same time, are regulated by some intrinsic mechanisms. This makes it challenging to link the competence state of RPCs to a specific developmental time, as well as to the fully directive action of external regulation. One of the underlying mechanisms of such stochasticity may be the extreme heterogeneity exhibited by RPCs in their expression of TFs [38]. Nevertheless, the selection of the fate of retinal cells is not completely stochastic, since the frequency of some, certain clone compositions, turns out to be higher than one could expect in the case of their completely stochastic development. Gomez and co-authors [33] support this point of view with a mathematical model, which shows that the probability of occurrence of early cell types, such as, in particular, ganglion cell precursors, decreases over time against the background of an increase in the probability of the production of late cell types such as rods, bipolar cells, etc., rather than disappearing completely.

The process of retinal self-organization includes not only the intriguing abilities of RPCs to behave within the framework of the competence model and, at the same time, stochastically, but also a number of other intriguing properties. One of them is the “overproduction” of cells of early cell populations. It has long been known that clones at early time points are usually much larger than later ones [16]. In later developmental phases, overproduction is purposefully eliminated through cell death so that the populations of retinal cell types reach a number adequate for their specific relationships and functions. Programmed cell death, or apoptosis, is known as a common phenomenon in embryonic development and cell differentiation during histogeneses, in particular in the CNS [39]. According to Vecino and Acera [40], in the case of the retina, the main functions of cell death are as follows: eliminating the neurons that have not established synapses with partner cells (targets); involvement in the retinal laminar structure formation; and eliminating the cells of transit populations. Cell death in the developing NR is described in a number of early studies, with their data summarized in a review by Valenciano et al. [41]. Two waves of cell death are reported for mammals: the first affects the RPC population in the early retinogenesis phase, while the second in the late phase, during synaptogenesis, when the plexiform layers of the NR are formed [40,41,42]. In addition to the above-mentioned phases, Valenciano et al. [41] indicate morphogenic cell death, a programmed cell death related to optic vesicle evagination, optic cup formation, and closure of the optic fissure. Many of the cellular and molecular mechanisms involved in cell death have been elucidated. In particular, many of the molecular triggers underlying programmed cell death have been discovered [43]. The survival/death of both RPCs and retinal neurons in the late stages of retinal development is regulated by families of molecules that carry out both internal and external control of these processes (summarized in [41,43]). Among external signals, Braunger et al. [43] distinguish NF and TGFβ; in the late phase, neurotrophins (BDNF, CNTF, and NF) and immune modulators are considered [41,44,45]. There is evidence of the role of cell death in the formation of cell mosaicism [10,46]. The resulting retinal mosaics constitute the “functional units” necessary for providing the normal functions of the retina [47,48].

The cell migration that occurs in the retina along with the processes of cell death and stratification and mosaicism formation aims to achieve the definitive localizations of cells in the NR structure [48,49]. The cell types that emerge after exiting the proliferative phase migrate along the radial (apico-basal) axes of the NR anlage. Tangential migration (perpendicular to the radial axis of the NR) is also known. It is reported to be characteristic of cells emerging in the early phase of retinogenesis. This allows them to move a short distance within their laminar position [48,49]. The processes of neuronal migration, lamination, and mosaicism formation are discussed in detail in a review by Amini et al. [46].

Spatiotemporal, precise orchestrated processes such as the fate choice (competence, differentiation) by cells, excess proliferation, migration, and death, as well as the definitive maturation of cells upon reaching the correct location (in terms of time and position) in the overall NR composition provide adult functionality of the retina. These processes are initially based on a high measure of plasticity of neuroepithelial cells to accommodate the spatiotemporal process while maintaining their tissue integrity and architecture. The self-regulation of plasticity properties against the background of intercellular and physical factors of influence in vivo is described in terms of the expression of genes, TFs, controlled by intrinsic and extrinsic factors of the cells.

### 3.2. Regulation of Eye and Retina Development by Extrinsic Factors in Vertebrates In Vivo

The role of intrinsic regulatory factors in response to external signaling systems can be observed at all stages of retinal development in vivo. A number of signaling regulatory molecules emitted from neighboring tissues are identified at the very early stage of eye development. A previous work implicates the extracellular matrix (ECM) as a major player in eye structures’ morphogenesis, with the mechanisms of this influence, however, being poorly understood and the roles of individual ECM proteins not fully defined [50,51]. The eye anlage is surrounded by periocular mesenchyme. However, the study of the effect of its ECM proteins is reported [52] to be complicated by the presence of two sources in the periocular mesenchyme: the neural crest and cells of mesodermal origin, whose specific role is difficult to identify. To understand the role of the neural crest, Bryan et al. [52] analyzed the proteins of its basal membrane. For this, embryos of mutant zebrafish lines were used, in which the basal membrane of the neural crest adjacent to the prospective RPE was destroyed genetically. Most neural crest cells were absent, resulting in optic vesicle cells that moved faster and farther than those in wild-type embryos. Rim movement was impaired in the absence of a complete, continuous basal membrane around the RPE, which resulted in optic cup malformations. A search for a key molecular effector involved in the interaction of the neural crest ECM with the developing eye indicated conserved sulfated monomeric glycoproteins, referred to as nidogens, acting as ECM modulators. The authors [52] suggested that the ECM of the basal membrane is a dynamic substrate capable of regulating cell movements in the early eye anlage, as well as spreading of its outer layer, the RPE. There is an opinion that the periocular mesenchyme, regardless of its source, controls the signaling pathways Hh, TGFβ, and Wnt regulating the eye development in vivo [51,53].

The role of the Wnt and BMP pathways in vertebrate eye development was assessed using inhibitors of these signalings, DKK1 and Noggin (for Wnt and BMP, respectively), and exposure to exogenous IGF1 (Glinka et al., 1997; Lamba et al., 2006) [54,55]. Other results [56] provide evidence that BMP signals in early development inhibit the acquisition of the eye field traits. However, at the stage of the formation of optic vesicles, BMP signals running from the forming lens, on the contrary, are necessary for the maintenance of the eye field character, inhibition of dorsal telencephalic cell identity, and specification of NR cells. The inhibition of WNT and BMP in the developing RPCs derived from ESCs in vitro [55] and the injection of IGF1 mRNA to clawed frog (*Xenopus*) embryos in vivo [57] induced retinal development. In the latter case, this occurred presumably due to the suppression of the Wnt signaling pathway by IGF [58].

The non-canonical, β-catenin-independent Wnt signaling pathway turned out to be important to form and maintain the eye field and to regulate cellular movements during morphogenesis. The development of the eye field has been shown to be at least partially controlled by Wnt11 and Fz5 through local antagonistic interactions with Wnt/β-catenin signaling, which suppresses retinal identity [59]. In turn, Wnt/β-catenin signaling plays an essential role in multiple developmental processes and has a profound effect on cell proliferation and cell fate determination. Faulty regulation of Wnt/β-catenin signaling results in multiple ocular malformations due to defects in the process of cell fate determination and differentiation [60].

In addition to Wnt, BMP, and IGF1, other factors should also be included in the system of regulation of the early retinal formation stages. It was found that FGF, TGFβ, Notch, Vax, retinoids, and Gas1 are responsible for the diversification and stabilization of the two major visual domains (RPE and NR) in the optic vesicles and eye cup [61,62]. The role of FGF and Wnt signaling has been studied in mice both during RPE and NR formation and in maintaining the properties of the retinal growth zone (ciliary margin (CM)). Using a single-cell analysis, Balasubramanian et al. [63] found that FGF along with Wnt signaling regulate the stem properties of CM cells and their entry into differentiation. FGF promotes Wnt signaling in the CM by stabilizing β-catenin, while Wnt signaling converts the NR into either the CM or the RPE depending on FGF signaling; FGF transforms the RPE to the NR or CM depending on Wnt activity. These data collectively showed that the vertebrate eye develops through a phase transition determined by a combinatorial code of FGF and Wnt signaling [63].

### 3.3. Regulation of Eye and Retina Development by Intrinsic Factors in Vertebrates In Vivo

The external signaling influences the expression of TFs, whose differential functioning ensures both early and advanced retinal development [11,12,64,65]. TFs are suggested to be the primary determinants of retinal fate choices (Figure 3 and Figure 4). The expression of genes responsible for the emergence of differences in the competence or, in other words, the potential of RPCs to produce certain cell types determines the processes of cell sequential diversification. It should be noted that TFs, whose differential expression is characteristic of the development of the eye and, in particular, the retina, are conserved across species, and this is certainly a convenience for assessing their role in the eye development mechanism [11,12,65].

As mentioned above, the eye’s development is initiated with the emergence of eye field cells in the anterior neural plate during late gastrulation. The formation of this region is accompanied and coordinated by the expression of the eye field TFs (EFTFs). The progenitor cells of the optic vesicle in this region express the same basic genes: *Pax6*, *Rx*, *Six3*,*6*, *Otx2*, *Sox2*, and *Lhx2*, encoding the respective TFs. In the anterior neural plate, EFTFs compose a genetic network to control the eye specification against the background of inhibition of BMP, Nodal, and Wnt/β-catenin signaling [65,66,67]. The work of TFs in their interaction with signaling pathways at different stages of the eye and retina development (RPE and NR) has been well characterized and presented in many works [11,29,30,64]. In brief, it can be represented as follows. As mentioned above, the presence of the eye field in the anterior part of the neural plate is characterized by the expression of EFTFs, in particular Rx. The latter is up-regulated due to the cooperative work of TFs Otx2 and Sox2. The interaction with the BMP, Wnt, and Shh signaling pathways leads to the separation of the eye field area from other areas of the developing brain. Then, the expression of Lhx2 occurs, which allows the initiation of the optic vesicle formation. The expression of TFs Otx2 and Mitf, induced by the TGFβ pathway, as well as Pax6 is characteristic of the optic vesicle in general. However, when the prospective NR is formed, Mitf is down-regulated with the involvement of Vsx2. The formation of RPE requires the suppression of FGF signaling, as well as Sox2, which occurs with the involvement of Otx2. Thus, the signaling involving simultaneously developing surrounding tissues leads to regionalization in the eye anlage, namely the formation of the RPE and NR domains [29,66,67,68]. The major transcriptional modulators to differentiate and maintain the specific RPE properties are Mitf, Otx, and β-catenin. However, after the determination of the domains, both cell populations retain the ability to convert one into another (RPE⟷NR). In birds, this is observed during embryonic development [69], as well as in the case of BMP application [70]. The RPE conversion into the NR is also reported for amphibians [71,72] (see below). In *Chx10*-null mutant mice, NR cells transdifferentiate into RPE cells against the background of developing microphthalmia [73]. 

As mentioned above, during the NR development, RPCs produce the full range of retinal cell types. They do not do this simultaneously, although frequently with overlapping. This indicates that, already in the early phase of NR genesis, RPCs represent a heterogeneous population that exhibits differential gene expression responsible for differences in the RPC potential to produce one or another cell type [74].

One family of TFs that has been shown to be most important in the regulation of cell fate is the basic–loop–helix (bHLH) family. The proneural bHLH transcriptional regulators are key components for the intrinsic programming of RPCs and are essential for the formation of the diverse retinal cell types [75,76]. In turn, the expression of key intrinsic regulators, multiple retinogenic bHLH. and homeodomain TFs, responsible for the specification of NR neurons, is controlled by the *Pax6* master gene [77,78].

Miesfeld et al. [79], using antibodies recognizing the Atoh7 (formerly Math5) polypeptide of mice and humans, as well as informative knockout and transgenic mouse tissues and overexpression experiments, found that the transient features of the Atoh7 protein and Atoh7 mRNA expression during retinal neurogenesis match the expected pattern on the tissue and cellular levels during the neurogenesis wave preceding the optic cup formation. The differentiational expression of TFs Olig2, Ngn2 (Neurog2), and Ascl1 (Mash1) is characteristic of RPC subpopulations [80,81,82]. Brzezinski et al. [83] attempted to link the heterogeneity of the RPC population with the differences of its cells in the expression of TFs-encoding genes. They used genetically modified mouse lines, their embryos, as well as retinal explants in vitro. Ascl1- and Ngn2-inducible expression fate mapping was conducted using the *CreER™/LoxP* system. It was found that RPCs represent a highly heterogeneous cell population expressing every possible combination of TFs. By summarizing the results, it became possible to divide the RPC population into at least two categories: early (Ngn2^+^), which gives rise to a population of ganglion cells, and late (Ascl1^+^), whose cells differentiate into other types of retinal neurons. Simultaneously, it has been shown that the DNA-binding protein, Ikzf1/Ikaros zinc finger TF, plays a role in determining the state of RPCs’ competence associated with generating early-born cell types. The inactivation of Ikzf1 caused a loss of early-born neurons including ganglion, amacrine, and horizontal cells without affecting late-born cell types [84].

In the last 25 years, a large number of TFs, combinations of genes, as well as cofactors responsible for the differentiation of specific NR cell types have been identified using molecular genetics approaches and bioinformatics. In particular, it was found that Atoh7 and Pou4f2 (homeobox) catalyze the rate-limiting step in the specification of retinal ganglion cells. Prox1 expression proved to be essential for the production of horizontal cells, while Neurod1 and Neurod4 and Pax6 and Six3 regulate the production of amacrine cells. Crx is regarded as a key TF for the specification of photoreceptors, while Vsx2 (Chx10) is key for the production of bipolars [85,86]. The expression of TFs Sox11 and Sox4, carried out in a coordinated manner during retinogenesis, is responsible for correcting the size of populations of certain specific cell types. As reported, the epigenetic mechanisms of gene regulation are used in this case [87].

The scope of the present review does not allow a detailed consideration of the dynamics of TFs’ expression during the maturation of each individual NR cell type. A more detailed description of the data found in this area of research is provided in earlier reviews, e.g., [25,30,31,88]. In many studies, a core transcriptional hierarchy underlying retinal cell types’ appearance in vertebrates is assumed. However, at the same time, in many studies, the complicated pattern of TFs’ expression is reported. It has been noted that TFs’ expression is influenced by simultaneous, synergistical expression of other TFs (combinatorial codes), which also plays an important role in cell fate diversification [89,90]. In RPCs, thousands of genes undergo differential expression, which are turned on or off as the major specific NR cell types arise. Although studies of the recent past have identified the key genetic regulators of retina specification, the question as to how their network works is still far from being fully addressed. In the review by Buono and Martinez-Morales [91], the authors confidently state that the system biology and the emerging techniques such as RNA-seq, ChIP-seq, ATAC-seq, or single-cell RNA-seq can make a significant contribution to understanding the operation principles of genetic regulatory networks in retinal development. Recently, Lyu et al. [88] used integrated single-cell RNA and single-cell ATAC sequencing (scATAC-seq) analysis and the models of developing mouse and human retinas to identify multiple interconnected, evolutionarily conserved genetic networks composed of cell-type-specific TFs that both activate genes within their own network and inhibit genes in other networks. It has been shown that such regulatory machinery can control temporal patterning in primary RPCs, regulate transition from primary to neurogenic progenitors, and drive the specification of each major retinal cell type. The authors exemplify this by TF nuclear factor I (NFI), which binds CAATT-boxes. It was indicated that this and other TFs selectively activate the expression of genes promoting late-stage temporal identity in primary retinal progenitors [88].

Along with studies of regulatory genetic networks, with the full complexity of this problem, attention is paid to epigenetic regulatory mechanisms, including the chromatin landscape, histone modifications, DNA methylation, non-encoded RNAs, etc. Thousands of enhancers are shown to be active in the developing retinae, and many of them have features of cell- and developmental stage-specific activity [23,62,92,93,94]. Attempts to identify the role of miRNAs were made earlier. MicroRNAs, single-stranded 19 to 25 nt small ncRNA that are part of the RNA-induced silencing complex (they pair with target sites located primarily within the 3′-untranslated region of mRNAs), are capable of suppressing gene expression by inhibiting the translation or causing degradation of RNA [95]. A study of the miRNAs profile by the application of the in situ hybridization method has shown that many of them are expressed both during retinogenesis and at the adult stage in overlapping and distinct patterns [96,97,98,99].

Norrie et al. [92] studied the dynamic changes in the 3D chromatin landscape by ultra-deep in situ Hi-C analysis on murine retinae during retinal development. Developmental stage-specific changes were identified in chromatin compartments and enhancer–promoter interactions. The authors designed a machine-learning-based algorithm to map euchromatin and heterochromatin domains’ genome-wide and overlaid it with chromatin compartments identified by Hi-C. Single-cell ATAC-seq and RNA-seq were integrated with Hi-C and previous ChIP-seq data to identify cell- and developmental-stage-specific super-enhancers. As a result, it became possible to identify the bipolar neuron-specific core regulatory circuit super-enhancers upstream of *Vsx2*, whose deletion in mice led to the loss of bipolar neurons [92]. A number of works consider the impact of the loss of polycomb repressive complex 2 (PRC2), which catalyzes the addition of the repressive mark H3K27me3, on retinal development. Mutations in the subunit of this complex (Ezh2, Ead) led to a noticeable decrease in the RPC proliferation, as well as to changes in the choice of the type of differentiation by progenitors [100,101].

Thus, studies considering the work of the genome and epigenome, the changes and modulations occurring in cells of the prospective retina as it self-organizes in vivo, being fundamental, are, nevertheless, still far from comprehensively addressing these issues. However, as can be seen, the prospects for such research have been outlined, and a technological capacity for complex molecular genetics and epigenetic studies has been built.

### 3.4. Morphogenetic Factors of Retinal Self-Organization In Vivo

The self-development of the retina implies the acquisition of its characteristic shape, the process of morphogenesis that occurs along with cell differentiation, and the formation of their coordinated functional circuit. In studying the role of TFs as morphogenetic factors, special attention is paid to *Pax6*, a master gene that is key to the control of eye development in both invertebrates and vertebrates [102,103]. In the work by Grocott et al. [104], who used chicks as a model organism, *Pax6* was shown to direct the expression of a pair of morphogen coding genes, *Fst* and *Tgfb2*, which modulate the *Pax6* function via positive and negative feedback. The topology of the *Pax6*/*Fst*/*Tgfb2* gene network proved to be consistent with the activator–inhibitor-type Turing network [105], which is capable of manifesting a self-organizing pattern-forming ability in the absence of position information. This process was computationally modeled, and the results indicated that the work of this genetic network is essential for establishing the primary axis of organization for the spontaneously polarizing retina and prefiguring its further development [104].

Nevertheless, the study of the TF expression, signaling pathways, and their regulatory networks does not seem to be sufficient enough to explain the mechanisms of self-organization and morphogenesis of the retina. In particular, it is not clear how individual cells determine the state in space and time and consistently modulate the formation of a 3D structure until the definitive shape that suits the function to be performed. As discussed below, the extensive reshaping of the eye primordium during development is highly conserved across vertebrates and is successfully reproduced in vitro in organoid models, using not only RPCs, but also embryonic (ESCs) or induced pluripotent stem cells (iPSCs) [106,107] (see below). This confirms the existence of the self-regulating properties of the RPCs and their ensembles involved in this process, which coordinate their actions by direct and indirect interactions between each other. The information about the molecular participants of these interactions in the early phases of self-organization is still insufficient [108].

Cellular mechanosensing mechanisms, which, apparently, should also be considered in the system of regulatory mechanisms of retinal morphogenesis, have been found using models of the development of other tissues [109,110]. The study of Okuda et al. [111] considers the mechanical aspect of the problem of eye and retina formation. The authors based their study on the previous data of experiments with the eye cups derived from mouse ESCs in vitro and the “relaxation–expansion” model proposed at that time [106,112]. In a study of cell displacements during optic cup formation, eye primordia of 9-day mouse embryos were used in ex vivo conditions. The key cell behaviors required for the invagination and the subsequent hinge formation along the NR–RPE boundary were identified. A conclusion was made that mechanical force plays a primary role in feeding back the 3D tissue deformation to the force generations of individual cells across different scales [111]. It is also known that mechanical deformations play a role in the formation of the optic fissure [113]. The Hippo signaling pathway, a main cell mechanotransducer that can respond to ECM stiffness, should also be mentioned in this context [114]. It is known that Yap and Taz, being the co-activators of the Hippo pathway, are involved in the RPE development as sensors of mechanical signals inside the cell nucleus [115]. Data obtained on a zebrafish model have shown that the Yap/Taz-Tead activity is necessary for a part of the RPCs, having equal potencies for both RPE and NR production in the optic vesicle, to acquire the RPE identity subsequently. The conclusion is supported by the Yap immunoreactivity in the nuclei of prospective RPE. Furthermore, zebrafish *yap* (*yap1*) mutants completely lost the population of RPE cells and/or exhibited NR colobomas (redundant proneural cell proliferation). These data allowed the conclusion that Yap and Taz are early key regulators of RPE genesis and one of the causes of congenital ocular defects [115]. The role of cell primary cilia, or rather the expression of genes regulating ciliogenesis, has recently been discovered in the morphogenesis of the eye. Fiora et al. [116] found that in *Arl13*-null mouse embryos, the lens is abnormally surrounded by an inverted optic cup whose RPE is oddly facing the surface ectoderm. It has been found also that *Arl13b* genes can modulate the work of the Shh signaling pathway along the dorsoventral (DV) axis and are thereby involved in setting the DV polarity and morphogenesis of the optic vesicle.

Many theoretical and practical attempts been made to show the role played by mechanics as a driver of tissue self-organization and morphogenesis [8]. Recently, attempts have been made to determine the role of RPE in the creation of the mechanical forces responsible for optic cup formation [117]. The authors managed to address this problem by genetically modifying the Tg (E1-*bhlhe40*:GFP) line of zebrafish. This allowed making all newly appearing RPE cells fluorescent. It was shown that, in the virtual absence of proliferation, RPE cells stretched and flattened, thereby matching the retinal curvature and promoting optic vesicle folding. The localized interference with the RPE cytoskeleton disrupted tissue stretching and optic vesicle folding.

The role of mechanical forces, as well as the ECM composition mentioned above, responsible for their generation, are traditionally discussed when considering the formation of the optic vesicle and the optic cup in vivo and in vitro (below), a well as in the development of congenital eye pathologies [118,119,120]. The ECM is involved in the regulation of the movements of individual cells and their groups; the binding of the ECM to the cell causes the contractile force to strengthen, with the subsequent transmission of this signal into the cell [118,121]. To describe the regulation of eye morphogenesis, when considering the role of mechanical forces and the ECM, authors often use mathematical models [104,112,118,122,123], as well as observations of changes in morphogenetic movements during the selective elimination of certain ECM proteins and their receptors [124,125]. Intraocular pressure (IOP), created through the accumulation of the aqueous humor of the eye, should also be mentioned among the physical forces that are factors influencing the formation of eye tissues in vivo. It has long been known that a slowdown/underdevelopment of the eye occurs with a decrease in IOP [126]. However, to date, there is no specific data on the role of flows of accumulated aqueous humor, the pressure they exert, and the resulting stresses in the tissue of the developing retina.

Proteins providing the planar cell polarity (PCP), i.e., a polarity axis that organizes cells in the plane of the tissue, are considered another morphogenetic regulatory factor. PCP proteins coordinate the planar polarity between cells and control polarized behavior by modulating the cytoskeleton [127]. According to Álvarez-Hernán et al. [128], the distribution of PCP proteins in the developing chick retina indicates their important role in the axonal guidance at early stages of retinogenesis, as well as possible involvement in the formation of cell asymmetry and the maintenance of retinal cell phenotypes. Thus, we see how works that are different, both in terms of goals set and methods used, appear in the field of studies of the physicochemical and mechanical factors of the regulation of eye and retina morphogenesis regulation. The methods include molecular genetics, mathematical simulation, measurement and computer analysis of cell displacements and deformations, etc. Against this background, interdisciplinary research and generalization of diverse information are required.

Thus, when considering the phenomenon of retinal self-organization during eye development in vivo, the multifactorial nature of the process, including both intrinsic and extrinsic regulatory mechanisms, should be taken into account. In brief, these are as follows: cell–cell relationships and intercellular signaling positive and negative feedback, expression of genes and TFs and their networks operating differentially in a spatiotemporal manner, the role of epigenetic regulatory mechanisms, regulation carried out by external signaling pathways, cell proliferation, migration, and apoptosis, changes in mechanical properties and cell-level forces that are regulated by molecular signals, etc. This list is a vivid demonstration of the biological complexity and genetic heterogeneity of the self-organization process. Data obtained by retinal regeneration research in vivo partially complement our knowledge of retinal self-organization.

## 4. Self-Organization of the Retina during Regeneration in Mature Amphibians

When considering the retinal self-organization issue, the epimorphic regeneration of the retina in mature amphibians attracts special attention. In the context of regulatory mechanisms of self-organization, the phenomenon is of interest because the retina develops de novo while being surrounded by differentiated, functioning tissues of the completely formed eye of matured animals.

In caudate amphibians (Urodela), which are the most regeneration-competent animals, the retina regenerates after the surgical removal of the NR or its death after cutting the optic nerve and blood vessels. In all types of eye surgery, the RPE and retained CM cells become the source of a new, complete retina [129,130,131,132,133,134]. The major events of the process are as follows: RPE cells’ withdrawal from the layer, the loss of their original phenotypic traits and properties (dedifferentiation), cell proliferation, the formation of an intermediate population of amplifying neuroblasts (Figure 5). Having reached a certain size of cell population, neuroblasts, representing the prospective NR, leave the reproduction cycle and acquire the phenotypes of retinal neurons and MG cells [129,130,131,132,133,134]. Furthermore, there are no disturbances of the NR layered structure, and its functional maturity has been shown [135]. The order of maturation of cell types in the regenerate matches that in normal development and is similar to the order found in other vertebrates [136]. The dynamics of the formation of the ONL, which is ahead of the INL in development, but lagging behind the ganglion layer, is described using the recoverin photoreceptor marker (Rec) [137] (Figure 5).

When the retina regenerates in caudate amphibians, RPE-derived source cells use the conserved molecular mechanisms that have been identified for the retina in the embryonic development of other vertebrates in vivo. The NR regeneration in Urodela is carried out under the control of TFs and signaling pathways. Native RPE cells in newts express melanogenic differentiation and specialization genes (*RPE65*, *Otx2*, *CRBP*). The expression pattern of developmental homeobox genes characteristic of the eye field in development (*Pax6*, *Prox1*, *Six3*, *Pitx1*, *Pitx2*), along with tissue-specific *RPE65* and *Otx2*, is described both at the beginning of the RPE cell-type switch and during the NR regeneration [133,138,139,140,141]. The up-regulation of *Pax6*, *Six3*, and the FGF2 growth factor genes has been shown to occur against the background of the suppression of *Otx2* expression and tissue-specific *RPE65* and *CRBP* [133,140,141]. The activity of immune response genes and proto-oncogenes *c-fos*, *c-myc*, and *c-jun* is detected in the early phase of RPE reprogramming, shortly after the disturbance of the normal interaction of the RPE and NR [142]. As was found in experiments on isolated cells using q-PCR, the first daughter RPE cells at the beginning of retinectomy-induced proliferation manifest the expression of pluripotency genes *c-Myc*, *Klf4*, and *Sox2* and, along with them, “developmental” *Mitf* and *Pax6* [143]. In the context of signaling cascades, studies of retinal regeneration in Urodela focused on FGF, BMP, Wnt, Shh, and Notch signaling [144,145,146,147]. As a study of the role of the Notch signaling pathway showed, the introduction of an inhibitor (DAPT) leads to premature maturation of neurons in the NR regenerate [144,145]. The major source of FGF2 signals is assumed to be the choroidal coat [148], CM and MG cells, and RPE cells proper [147]. Thus, the conducted studies have revealed similarities of the range of expressed regulatory molecules, TFs, and signaling pathways with those in the development of the vertebrate retina.

The study by Kaneko et al. [149] described the dynamics of programmed cell death (apoptosis) in retinal regenerate cells in situ following the ablation of the original NR. The first wave of apoptosis was observed at the stage of active proliferation of RPE-derived progenitor cells. The second wave, affecting the cells of the INL and the ganglion layer, took place later, when the NR underwent stratification and plexiform layers formation. Thus, as in the case of in vivo retinal development in other vertebrates [40,42], cell death during NR regeneration mandatorily accompanies the main phases of NR self-organization and histogenesis.

All stages of the retinal regenerate formation in the newt occur against the background of ubiquitous expression of stathmin, a small-sized cytoplasmic phosphoprotein known to be a microtubule regulator. As is known, the reorganization of microtubules can influence the shapes of individual epithelial cells, as well as the shape of whole tissues [150]. Stathmin expression is observed both during the formation of RPE-derived NR regenerate and during stratification, the formation of plexiform layers, and synaptogenesis. Subsequently, these proteins, which are intracellular regulators of cytoskeleton microtubules, participate in maintaining the structure of the formed NR regenerate [151]. This aspect of retinal self-organization, in particular the role of cytoskeletal components and their extracellular and intracellular regulators, is also of interest with respect to other examples of retinal self-organization and in the context of speculations about the relationship of the cytoskeleton with dynamically changing mechanical and adhesive events in the retinal anlage.

When discussing the self-organization of the retina during its regeneration in an “adult” surrounding, it is worth mentioning the cases of the development of stratified histotypic NR structures after the implantation of RPE fragments into the cavity of the lensectomized eye of newt [148,152] and frog *X. laevis* [153,154]. When RPE layer fragments together with underlying choroidal and scleral coats were transplanted into the posterior eye chamber, they regenerated the NR in a similar manner as they do in the regeneration process in situ. In this case, RPE-derived, self-organizing NR, as in the case of in situ regeneration, was a target of proteins produced and released by mature eye tissues (including intact retina) [155]. One of them is the FGF2 ligand. This factor and its receptors FGFR2 were found in various tissues of the newt eye, including the retina [147]. The choroid underlining the RPE is also considered a source of FGF2 [148]. It is assumed [148] that FGF2 coupled with IGF-1 influences the formation of NR regenerate in the eye.

The phenomenon of retinal self-organization in the surroundings of the tissues of the completely formed eye can be explained by (1) the RPE cells’ plasticity and (2) the competence of RPE-derived RPCs to form neural cell types and MG [156,157]. In this case, it is possible to assume some similarity of extrinsic signaling factors acting in the development and in the adult state in newts. This assumption is based on the fact of the paedomorphic status of newts, which retain a number of juvenile traits upon reaching maturity [158,159,160].

Retinal regeneration in vivo was also found in 1–8-month post-metamorphic frogs, *X. laevis* [72,161,162]. In these animals, the retinal regeneration/self-organization occurs after the removal of the original NR, provided that its vascular membrane is retained. In this case, RPE cells leave the layer, migrate, settle on the membrane, dedifferentiate, and form a population of proliferating neuroblasts, the NR anlage, which then develops into a stratified retina, similar to a normal one. In this case, NR self-organization occurs not only in the surroundings of the tissues of the mature frog eye, but also in an atypical localization, away from its cell source (RPE) and other tissues of the posterior wall of the eye. The vascular membrane is laminin-immunoreactive. A culture study has indicated that this and other ECM components are needed for RPE cell reprogramming and the formation of a retinal stratified structure [161]. The key regulators of NR regeneration in frogs, as well as TFs and signaling pathways have been investigated. In particular, FGF2 has been shown to accelerate RPE transdifferentiation in vitro and in vivo and is necessary to maintain Pax6 expression [154,162].

Thus, the use of amphibian animal models has shown that mature animals demonstrating NR regeneration due to RPE cells’ reprogramming have a similarity in the molecular and cellular events accompanying the process with those working in the normal development of the vertebrate retina. Similarities are found both in the expression of conserved, “developmental” TFs, key signaling pathways, in the dynamics of cell migration and death, as well as in the order of maturation of the cell types. Hence, the capability of the self-formation of RPE-derived retinal anlage in mature amphibians in situ, as well as that in the eye cavity after transplantation is based on the plasticity of the RPE and the competence of RPE-derived progenies (RPCs) to differentiate into specific retinal cell types and to self-organize in the population. These models of retinal regeneration can successfully be used for the further study of the retinal self-organization phenomenon in vivo under the influence of the “adult” surrounding. This capability is important, since non-hereditary retinal damages occur more frequently in an adult organism and should be subject to correction also in it. The advantages of these models also consist of the good knowledge of the phenomenon, the knowledge of the genomes of model newt and frog species, the low rate of the regeneration process (over 1 month), and large cell sizes with their smaller number in the populations of specific NR cell types [160].

## 5. Retinal Self-Organization In Vitro

### 5.1. Embryonic Retina Self-Organization in Reaggregation Culture

As is known, various cell types after dissociation are capable of reaggregation, self-organization, and tissue formation in the absence of inducing factors and tissue scaffolds. Studies on monotypic cells of sponges and sea urchins provided early evidence of the phenomenon [163,164]. In the studies by Moscona and co-authors [165,166,167], cells of complex vertebrate tissues, in particular the chick retina, were found to be also capable of aggregation, self-organization, and histogenesis. Layer and co-authors found a way to obtain fully stratified retinal aggregates (histotypic “retinospheroids”) from embryonic retina (ER) cells of chicks in a rotary culture [168,169,170,171,172,173,174]. Cells from different ER regions and obtained from chicks of different developmental periods were used in reaggregation cultures. When the CM was retained along with the adjacent chick RPE (E9), a prolonged proliferation and differentiation of CM cells were observed in the emerging retinospheroids, which indicated higher growth potentials of CM cells that exit the proliferation phase at stage E4 in vivo [175].

Some works [169,171,173] considered the role of embryonic RPE in the histogenesis of retinospheroids, and it was found that the correct layered organization, the growth of MG cell processes, and the formation of plexiform layers depend on the presence of these cells. Further studies [174,176,177] determined the inductive and decisive role of the RPE for retinal genesis and showed that the RPE effect is not independent, but often coupled with that of MG cells. In experiments with retina reconstruction on cell reaggregates from neonatal gerbil retinae, it was found that the RPE is a producer of factors promoting the formation of almost complete retinal spheres. The neurospheres maintained in the RPE-conditioned medium had an improved lamination [177]. It should be noted that the presence of the RPE is required for the normal morphogenesis of the mammalian eye in vivo. Early studies suggest that the RPE contact is necessary for the development and survival of photoreceptors [178]. The genetic ablation of the RPE carried out on a mouse model resulted in the destruction of the laminar NR organization [179].

The study of reaggregates derived through the cultivation of embryonic (E6) chicken retina cells has identified the role of MG cell processes in the organization of the glial scaffold for IPL establishment, as well as in the regulation of the process by cholinergic starburst amacrine cells and by L-glutamate [180]. As is known, the in vivo differentiation and regenerative potencies of MG depend on the activity of Notch signaling, which is known to drive cell–cell communication in the retina during development, damage, and recovery [181,182,183]. The inhibition of the Notch pathway in zebrafish at 45–48 hpf can block MG development in vivo. By decreasing the Notch levels with the γ-secretase inhibitor DAPT in the embryo media immediately before 45 hpf, it became possible to show that the process of self-organization is still implemented and the retina is formed in the absence of MG. However, it turns out to be less resistant to tension, much softer than controls, and had a strong tendency to rip apart. These data indicate both the existence of compensatory mechanisms in the ER self-organization in the absence of MG and the important structural/mechanical role of MG cells [182]. The use of the mouse ER in vivo and in vitro showed that blocking the development of MG by using either a BMP receptor antagonist or noggin leads to the permanent disruption of the retina, including defects in the outer limiting membrane, rosette formation, and a reduction in functional acuity [184]. These studies are evidence of the essential role that the maturing MG population plays in the retina formation both in vivo and in the conditions of reaggregation cultures in vitro. Therefore, the data provided in this section indicate an important, even leading role of maturing RPE and MG cells in the self-organization of retinal reaggregated cultures in vitro, as well as during retinal histogenesis in vivo.

Currently, molecular genetics studies are used to investigate the phenomenon of retinal self-organization after dissociation and reaggregation of ER cells. The zebrafish model and its mutant lines are an example of the application of new opportunities. Thus, studies are carried out on the Spectrum of Fates (SoFa1) line, in which the main differentiating cell types of the retina are labeled with fluorescent proteins driven by fate-specific promoters [185]. After dissociation, ER cells aggregate in vitro for 2 days and then self-organize into a layered structure for another day [186]. The use of this model makes it possible to study the mechanisms of cell–cell interactions and qualitative and quantitative observations. In the work of Eldred et al. [37], the SoFa1 model was applied to understand the significance of MG and RPE cells in retinal lamination. An image analysis allowed deriving the quantitative measures of lamination. As a result, the authors [37] concluded that MG is most essential for this process, which was consistent with the results obtained on cultures of chick reaggregated ER cells [187]. Thus, observations of the role of RPE and MG cell populations in the reaggregation cultures of chick and zebrafish ER cells support the idea of the leading role of some organizer-like signaling centers in the self-organization of developing/regenerating tissues, as well as about the cell–cell communication via chemical and mechanical feedbacks [7].

Assumptions about the role of the RPE and MG in the context of their involvement in the production of self-organization physical factors are no less interesting. As mentioned above, the maintenance of the retinal reaggregate architecture and tension depends on MG cells [182,188,189]. Tensile and stretch forces are presumably lost when embryonic MG cells are dissociated, but are re-established as the cells finally differentiate in retinal reaggregates.

Like the MG cell population, the RPE is a source of not only a number of chemical factors, but also physical effects in retinal 3D reaggregates. As for other self-organizing tissues, the differential adhesion hypothesis, which assumes the strongest adhesion between RPE cells, applies to aggregation cultures of ER cells. The next strongest one is suggested to exist between photoreceptors and/or bipolar cells, which can occupy the center of aggregates when the RPE is removed. These possibilities were tested using new advances in atomic force microscopy and micro-physical measurements of tension and adhesion [182,190,191].

When considering the issue of production of chemical factors involved in the ER self-organization process, the role of retinospheroids’ anterior rim, which produce molecules controlling lamination, is suggested. Nakagawa and colleagues [192], using chicken retinal reaggregation cultures, have found that the anterior rim is a source of a signal to rearrange the rosette-forming cells into a neuroepithelial structure. This activity was neutralized by a soluble form of Frizzled, which works as a Wnt antagonist. The neuroepithelial structure induced by Wnt-2b subsequently developed into correctly laminated retinal layers. The authors [192] hypothesized that the anterior rim acts as a layer-organizing center in the retina by producing Wnt-2b. The study of the factors, possible regulators of neurogenesis, and differentiation of photoreceptors in retinospheroids indicated the cooperative action of PEGF and GDNF as having the best effect on the differentiation of photosensitive cells [193].

Using the original protocol of neonatal mouse RPCs’ cultivation (heterochronic pellet assay), an attempt was made to assess the influence of mature retinal neurons on the production of differentiating progenitors [194]. For this, early-stage ER cells were dissociated, BrdU labeled, and mixed with a 20-fold excess of dissociated differentiated NR cells in vitro. Such combined cell reaggregates were cultured as a pellet on a membrane in vitro, and the type of BrdU^+^ proliferating cells was identified. As a result, a negative effect of mature cells of one or another differentiation type on the further production of new cells of the same phenotype was detected. Additionally, it was found that Shh is a negative regulator of ganglion cell maturation (the cell type specializing first) capable of prolonging the proliferation. It was noted that this approach can be adapted to other lineages and tissues to assess cell–cell interactions between two different cell types (heterotypic) in either an isochronic or heterochronic manner [194].

Thus, by using the method of dissociation followed by reaggregation of fish, chick, and mouse ER cells in vitro, it became possible to obtain information indicating an important and multifunctional role of maturing RPE and MG cells in the self-organization of the retina. These populations can be regarded as a kind of retinal self-organization center that regulates cell communication via chemical or mechanical feedbacks. Simultaneously, the regional hierarchy of regulatory influence in retinal histotypic reaggregates is not ruled out, as well as the mutual influence of cell populations having different maturation timings.

In the last two decades, retinal self-organization research has been further developed and showed a broad perspective for the translation of the results into biomedicine. This happened as a result of the transition from the production of organized histotypic 2D and 3D reaggregates to the creation of retinal organoids.

### 5.2. Retinal Self-Organization in the Process of Organoid Formation In Vitro

The previously established basic ideas about the development of the retina and its regulation factors in vivo, along with data from studies of 2D cultivation of ER, as well as in vitro self-organizing histotypic 3D cell aggregates became the basis for deriving retinal organoids from pluripotent stem cells (PSCs). The development of this area was simultaneously promoted by the extensive study of PSCs and the designing of 3D cultivation methods (Figure 6).

The modern term “*organoid*” refers to cells growing in a defined three-dimensional (3D) environment in vitro to form mini-clusters of cells that self-organize and differentiate into functional cell types, recapitulating the structure and function of an organ in vivo [195]. The experiments of the team headed by Y. Sasai [106,107,112,122] became the pioneering works in the creation and study of retinal organoids. These studies showed that neither dynamically changing in vivo conditions, nor a specific cell source, committed ER cells, are necessary for the development of an organized 3D retinal structure. The authors managed to derive self-organizing retinal stratified structures similar to in vivo-developing retina from embryonic stem cells of mice (mESCs) [106,196] and humans (hESCs) [107] and described them. A method of a serum-free floating culture of embryoid body-like aggregates (SFEB) and conditions enabling the self-organization of retinal lamella with the addition of Matrigel matrix were used for this. The presence of exogenous components (beyond media), nodal (TGFβ ligand), and ECM components laminin-1 and nidogen, required to elicit the optic cup morphogenesis, was also provided by the research protocol [106]. The addition of fetal bovine serum and the hedgehog agonist smoothened agonist (SAG) augmented the retinal differentiation of human stem cells with laminated retinas, expressing markers of all retinal cell types [107]. The human retinal organoids were larger than mouse organoids and had the ability to grow into multilayered tissue containing both rods and cones [107]. When adequate conditions were created in vitro, cultured cells began to develop in the neuroepithelial direction. The neuroepithelial anlage arose spontaneously and formed hemispherical epithelial vesicles, which became patterned along their proximal–distal axis. Its proximal part differentiated into the layer of RPE cells, while the distal portion (prospective NR) folded inward. As a result, a structure resembling an embryonic optic cup arose, where the stratified NR tissue was formed through interkinetic nuclear migration, with differentiating interneurons and photoreceptor cells [106,107]. The description of morphogenesis included four phases: (1) the uniform contraction of the apical surface produces a spherical vesicle; (2) regional relaxation creates a placode; (3) apical constriction at the placode border causes local cell wedging and slight invagination; (4) the growth of the placode region deepens the invagination. To interpret the tissue dynamics enabling the spontaneous invagination of the NR, the “relaxation–expansion” model was built, which includes three consecutive local events (relaxation, apical constriction, and expansion) [112]. The results indicated that in organoids, as under in vivo conditions, a complex system of cell–cell relationships and mutual influences that determine self-organization develops already at the first stages. From a mechanical point of view, this stepwise process proceeds autonomously without external forces from the surroundings such as lens placode and periocular mesenchyme [106].

The first studies using human pluripotent stem cells (hPSCs) to derive retinal organoids opened the prospects for further, large-scale production of 3D “mini-retinas” (organoids) [197,198,199,200,201,202,203,204]. Retinal 3D hPSC-derived organoids offered new opportunities to study the mechanisms of retinal degeneration and provided new models for drug discovery and cell-based therapeutics. The major issue of ensuring retinal histogenesis from hESCs, as well as from hiPSCs in vitro was the maintenance of the necessary conditions for a rather long time. For this purpose, bioreactors were used, which allowed an increase in the cultivation time and analysis of cellular, molecular, and biochemical processes in organoids with strict monitoring of physical and chemical conditions (temperature, pH, composition of the medium, oxygen level, etc.) [205,206]. Bioreactors and technologies for deriving 3D retinal organoids are constantly being improved. The major issue of providing retinal histogenesis from hESCs, as well as from hiPSCs, was the maintenance of the necessary conditions for a considerable period [207,208,209,210,211]. They should include the availability of physical and chemical factors necessary for self-organization and contribute to cell–cell interactions, organoids’ morphogenesis, and viability. Reproducing the distribution gradient of various soluble factors is also important. Gradients of soluble biomolecules, as well as nutrients and oxygen in the microenvironment affect stem cell differentiation and morphogenesis [212,213].

Retinal organoid studies take into account the information on the expression of regulatory molecules and data on the key signaling pathways and TF expression in vivo. It has been shown that, in early retinogenesis, hPSC-derived optic vesicles co-express *VSX2* (the earliest specific NR marker) and microphthalmia-associated transcription factor (*MITF*), whose expression was detected in the NR and RPE, respectively [214,215]. It has also been reported that WNT signaling is involved in the fate choice (NR/RPE) of cells of hiPSC-derived early organoids. Capowski et al. [216] performed VSX2 ChIP-seq and ChIP-PCR assays on early-stage optic-vesicle-like structures. WNT pathway genes appeared to be direct regulatory targets of VSX2, which suggests that VSX2 may act to maintain NR identity due to the RPE, in part, by direct repression of WNT pathway constituents. Earlier, the case of 2D cultures produced from hPSC-derived progenitors treated with Noggin/Dkk1 showed that, when exposed to a combined inhibition of BMP and Wnt and the positive effect of IGF-1, the genes characteristic of RPCs are expressed in cells [55]. Evolving from that study, the first description of hPSC-derived organoids employed Noggin and DKK1 to induce neural induction [198]. Other guided approaches incorporated IGF1, BMP4, or WNT antagonists to induce the NR as well [201,217,218]. It has also been found that the growth factors FGF3, FGF8, FGF9, and FGF19 are expressed in hESC-derived organoids at high levels [215]. The results obtained by Gamm et al. [219] have additionally shown that FGF9 acts in concert with VSX2 to promote NR differentiation in hiPSC-derived optic vesicles. A detailed analysis of the data on the activity of signal molecules, in particular FGF, in in the early hiPSC-derived retinal organoids allowed O’Hara and Gonzalez-Cordero (2020) to suggest the greater complexity of regulation by signaling factors than was initially deciphered from classical studies [220].

A transcriptomic analysis has identified the plasticity in the existence of cell cluster transition zones in the early postmitotic cell fate specification between progenitors and differentiated neurons [221,222,223]. In the study by Sridhar et al. [223], a comparative analysis identified similarities and differences between hiPSC-derived retinal organoids and human ER. Despite the disturbances in the layered organization, the organoids largely repeated the ER in terms of development timing and cell composition. However, under the same cultivation conditions, ER fragments formed 3D organoids with more advanced layered organization and synaptogenesis. ER-derived structures contained ACs, HCs, and BCs, as well as demonstrated the development of the OPL and IPL. In a study of transcriptomes of individual cells using scRNAseq, differences in expression levels of some cell-type-specific genes were noted [223].

Other comparative studies of organoids derived from PSCs and ER progenitors have also revealed a high degree of structural and molecular genetics homology [224,225,226]. At the same time, new expression profiles of the genes responsible for the competence of RPCs have been identified in cells of hESC-derived organoids [227]. Using scRNAseq, the authors found two distinctive RPC subtypes having different, multipotent and neurogenic molecular profiles. It has also been found that genes related to the Notch and Wnt signaling pathways, as well as chromatin remodeling are subject to dynamic regulation during the RPCs’ commitment [227]. The differences found in the structure of the developing retina and the levels of gene expression are explained by the conditions of cultivation and the lack of in vitro interstitial interactions [223]. The latter is often emphasized in the literature [228,229,230]. The lack of the RPE and lens has been noted to be a cause of changes in the effects of extrinsic factors and morphogen gradients [231,232]. The imperfect RPE morphogenesis in organoids is manifested as the formation of RPE patches instead of the layer normally underlying the NR and interacting with photoreceptors. To address the problem, a suggestion has been made to create additional physical conditions at an early stage of optic cup formation [233] (see below). The absence of the choroidal coat in organoids results in a deficiency of Indian hedgehog (Ihh) signaling, which is in vivo emitted from endothelial cells and required for the development of the RPE and scleral coat in mice [231]. It is also noted that, since organoids lack a blood supply, they have poorly access to various factors introduced to the environment [234].

There may also be a lack of retinoic acid (RA) signaling function regulated in vivo by the developing lens [232] and necessary for the maturation and function of photoreceptors [204]. Reconstructing conditions for the phototransduction process in organoids, which in vivo is formed through a complex cascade of gene regulatory networks in a strictly ordered retinal structure, is largely a challenge. Furthermore, there are approaches that accelerate photoreceptors’ maturation by using RA treatment in the pleiotropic all-trans-retinal form, whereas opsin-specific 9-cis-retinal enhances rod generation, and hypoxia facilitates cell survival [199,235,236]. Changes/disturbances of BMP/TGFβ signaling, in vivo controlled by ocular surface ectoderm-derived Smad4, are also probable. It is known that the loss of Smad4 leads to the microphthalmia and dysplasia of the retina in vivo [237].

The formed retinal organoids lack a ganglion layer, which is their significant disadvantage in terms of application in biotechnological research and regenerative medicine. It has been shown that young retinal organoids derived from hESCs and hiPSCs are similar to the developing human fetal retina [197,201]. They have ganglion cells, which are known to mature early in vivo [22,238]. It was shown long ago that ganglion cell precursors cultured in isolation are capable of recapitulating complex dendritic branching patterns [239,240], which indicates a competence dictating the dendritic morphology and cell specification in these cells. Ganglion cells isolated from early retinal organoid cultures contain diverse retinal ganglion cell (RGC) expression profiles and exhibit a divergent expression of guidance receptor genes [241]. With prolonged cultivation of organoids, in the absence of a target for axon growth, a visual center, ganglion cells are stochastically and progressively lost [199,242]. Following this, some interneurons are also lost, synaptic connections are remodeled, and stratification is partially lost [220]. The problems associated with the death of ganglion cells in organoids are suggested to be addressed by developing an assay for RGC survival-promoting molecules. One approach demonstrated that early in development, BMP4 can increase the number of early maturing RAX^+^ cells in 3D retinas [217]. RGC neurite outgrowth is promoted via substrate modulation, with laminin introduced to increase the outgrowth length and netrin1 for growth cone extension [241].

It is also clear that exogeneous neurotrophic factors are important. Growth factors belonging to the neurotrophin family, such as BDNF, can regulate retinal ganglion cell arborizations and survival [242]. However, it is evident that an artificially created surrounding cannot completely substitute the natural one due to the complex molecular context of retinal development in space and time in vivo.

Late-maturing Müller cells behave in a special manner in retinal organoids derived from hPSCs. Müller cells are preserved in the inner portion of the retina, avoid death, and are capable of establishing contacts with the cells surrounding them. A scRNAseq study revealed a large number of cells identified as Müller glia in late-stage organoids (>26 weeks) [223]. An interesting observation, also made using scRNAseq, indicates that the Day 90 retinal-organoid-derived Müller glia and photoreceptors cluster have shared transcriptional profiles [221]. This may reflect the molecular basis of origin from a common precursor, as well as explain the well-known ability of Müller cells to convert into a photoreceptor phenotype in vivo [243].

Microglia are resident tissue-specific macrophages and play a critical role in retinal development and homeostasis [9,10]. Recently, Gao et al. [244] provided an efficient procedure to generate microglia from hESCs. When co-cultured with human retinal organoids, hESC-derived microglial cells differentiated into retina-resident microglia, integrated with a typically ramified morphology, and demonstrated correct localization and proper function. These results present a new toolkit of “integral retinal organs” to study retinal microglial biology, retinal development, and retinal regenerative diseases [244].

A specific feature of the development of 3D retinal organoids in vitro is its duration. A transcriptome analysis of retinal organoids has shown that they reach a stable developmental state by Weeks 30–38 [245]. In the first instance, this is a convenience for studying slowly maturing cell types, the work of molecular regulators of the process, as well as the slowly forming phototransduction process. In the second instance, the duration of cultivation requires long-term availability of conditions necessary not only to maintain the viability of the tissue, but also to provide its development in a spatiotemporal way. Observations have been published that the process of the maturation of retinal organoids occurs faster under hypoxia conditions, which are known to exist during eye development in vivo [246].

The role of physical factors in the process of retinal self-organization during in vivo development is discussed above. When retinal organoids are formed, remodeling of intercellular adhesions, coordinated cell movements, and the regulation of cell surface tension can contribute to the retinal patterning as well. The mechanobiology of retinal organoids is currently an understudied field. Previously, bio-mechanical rules were specified that work with respect to both mPSC-derived optic cup invagination and hPSC-derived optic vesicles. These include inhibition of myosin, motor proteins that mediate folding, wedge shaping of the hinge epithelium, and the NR folding mechanism referred to as tangential expansion [107]. As is known, contraction of the actin-myosin cytoskeleton is a critical effector of hESC death, and the disruption of contraction by the inhibition of Rho-associated kinase (ROCK) or myosin light chain kinase can increase cell viability [247]. Therefore, the application of the ROCK inhibitor in the differentiation methods adopting dissociation–reaggregation is a common approach [107,248]. The study by Lowe et al. [233] showed that intercellular-adhesion-dependent cell survival and ROCK-regulated actomyosin-driven forces are highly essential for the self-organization of retinal organoids. These results have supported the hypothesis that newly specified VSX2^+^ RPCs form characteristic structures in equilibrium via minimization of cell surface tension [233].

The development of technologies for the production and study of retinal organoids pursues several goals (Figure 6). Much of the current research addresses the issue of the use of human retinal organoids for therapeutic and clinical implementation. One of the objectives is to understand, using organoids, the causes and mechanisms of the most common retinal disorders such as age-related macular degeneration (AMD), retinitis pigmentosa (RP), and glaucoma. Retinal organoids allow drug screening and disease modeling, as well as combining them with new genome editing tools. Some perspective and major goals of this work are considered in recent publications [229,230,249]. A special field is the use of hiPSC-derived organoids of the retina suffering from one or another genetic retinal disorder (patient-specific organoids) [250,251,252].

The great similarity of the hPSC-derived organoids with the retina of human embryo provides an opportunity for the modern study of the development of the human eye, an object inaccessible for experiments. Omics datasets combined with the advanced culture methodology can be applied to retinal organoid models, which will help understand the intricate process of human retinogenesis and retinal disease [220]. Retinal organoids are discussed as a source of cells, including young photoreceptors, as well as NR fragments that may be transplanted and compensate for cell deficit in the diseased, degenerating the retina [230,253,254]. The experience of deriving retinal organoids from PSCs should facilitate attempts to grow organoids from adult mammal retina cells, which are the retinal regenerative reserve. RPE cells, ciliary body cells, and Müller glia cells of mammals and human are known for the ability to form neurospheres containing cells of neural/retinal differentiation in vitro [161,255].

## 6. Conclusions

“Self-organization” is a broad concept that applies to biological, physical, chemical, cognitive, and social systems. Self-organization underlies the development of the eye retina and other tissues in embryonic ontogeny. In the present review, this phenomenon was described for the cases of retina development in vivo, during retina regeneration in situ, in the eye cavity in vivo, as well as under in vitro conditions. In vitro self-organization is observed in both 2D and 3D reaggregation cultures derived from embryonic RPCs and during the formation of retinal organoids from mammalian and human PSCs. Retinal self-organization is a highly intricate process and is far from being completely investigated. Self-organization is based on intercellular interactions: cell–cell communication using positive and negative feedback, selective adhesion, cooperative movements, and cell rearrangement, with mutual regulation of behavior in the retinal anlage. The molecular basis of the processes is the differential expression of genes, TFs and their targets, and changes in the epigenetic landscape, which, in turn, are regulated by intercellular signaling in a spatiotemporal context. The cells involved in retinal self-organization (RPCs and PSCs) exhibit the competence to differentiate into the cell types of the retina. These potencies are consistently modulated and implemented in the space and time of retinal formation. The molecular basis of the implementation of the competence is described in terms of gene expression and regulatory networks controlling it. During retina development in vivo, the process of self-organization is accompanied by the interactions with the surrounding developing tissues. These interactions are also based on complex regulatory molecular networks. The identification of the physical factors of influence and mechanotransduction, changes in the cytoskeleton, selective cell death, regulation of cell migration, and identification of cellular mosaicism are no less important components of the study of the retinal self-organization phenomenon. Thus, the construction of mathematical models, computational simulation, the use of animal models with genetic labeling of cells, advances in atomic force microscopy, micro-physical measurements of tension and adhesion, etc., are relevant approaches for this purpose.

To date, the self-organization phenomenon has been put forward as a basis for the production of retinal organoids, a promising model for a wide range of applications. With the use of omics datasets, combined with the advanced culture methodology based on retinal organoids, it has become possible to address the following issues: the study of the fine mechanisms of retinogenesis and retinal disorders, testing of pharmaceuticals, obtaining cells for transplantation, and the use of gene therapy for the treatment of hereditary diseases.

The process of self-organization still requires further integrated understanding, new molecular genetics approaches, multiple next-generation “omics” technologies, mathematical modeling, and interdisciplinary research. One of the most important and challenging objectives in the study of retinal self-organization is to collect information about cell–cell communication via chemical or mechanical feedbacks and about intracellular changes in the RPC population induced by them at the early stages of the process. Considered comprehensively/together, the examples of retinal self-organization described in the paper can shed light on the interdependence between cellular gene expression dynamics and cell–cell communication as a common theme of self-organization during development.

## Figures and Tables

**Figure 1 biomedicines-10-01458-f001:**
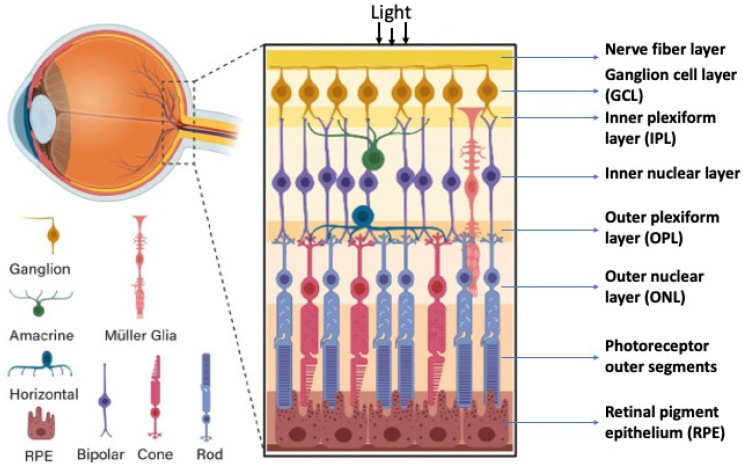
The structure of the retina and its main cell types. Schematic diagram (BioRender, https://app.biorender.com/).

**Figure 2 biomedicines-10-01458-f002:**
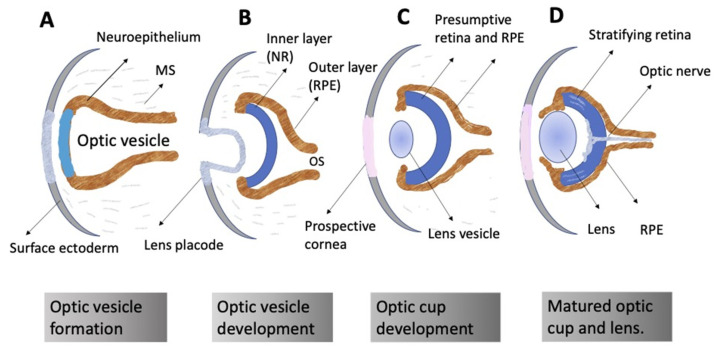
Schematic representation of vertebrate eye development. (**A**) Formation of the optic vesicle. (**B**) Specification of the NR (inner layer), RPE (outer layer), and optic stalk (OS) domains within the optic vesicle. Formation of the lens placode from the surface ectoderm. (**C**) Formation of the optic cup and the lens vesicle. (**D**) Organization of the formed eye. Neural retina stratification; growth of the optic nerve. MS: mesenchyme.

**Figure 3 biomedicines-10-01458-f003:**
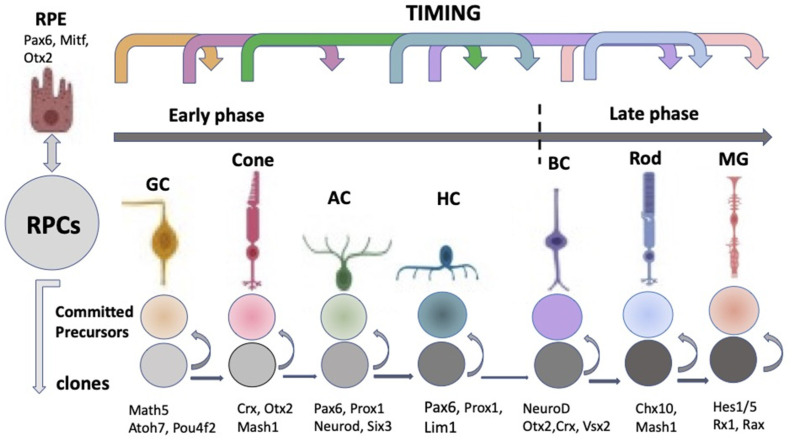
Schematic representation of the classical competence model for eye development. According to the model, retinal progenitor cells progress through the competence windows during which a specific retinal cell type is generated. RPCs—retinal progenitor cells; RPE—retinal pigment epithelium; GC—ganglion cell; AC—amacrine cells; HC—horizontal cells; BC—bipolar cells; MG—Müller glial cell. At the top: the sequential manner of a retinal cell’s differentiation. It has early and late phases. The represented sequence is general for vertebrates, although conditional, since the stages overlap with each other and have species-specific features. Key transcription factors responsible for retinal cell type specification are indicated at the bottom.

**Figure 4 biomedicines-10-01458-f004:**
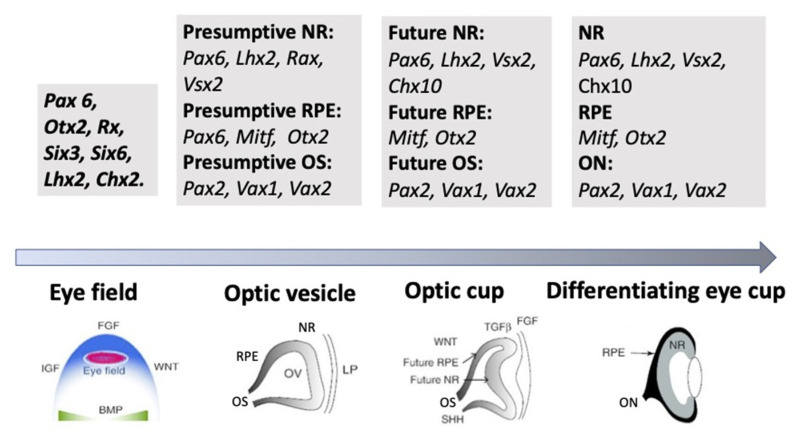
A schematic illustration of the groups of key transcription factors whose expression is associated with the main successive stages of eye development. NR—neural retina, RPE—retinal pigmented epithelium, OS—optic stalk, ON—optic nerve. See the detailed description is in the text.

**Figure 5 biomedicines-10-01458-f005:**
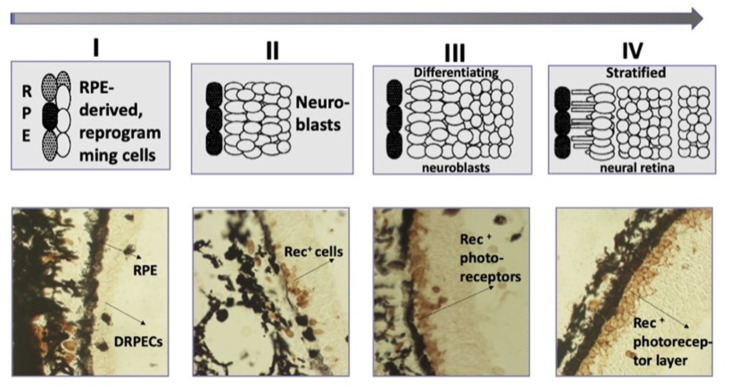
Regeneration of the retina in caudate amphibians. Above: (**I**)—the exit of RPE cells from the layer, the loss of initial traits and properties (dedifferentiation); (**II**)—the formation of an intermediate population of amplifying neuroblasts; (**III**)—NR anlage cells exit the cycle of reproduction having accumulated a certain number of their population; (**IV**)—NR anlage cells acquire the retinal neuron and glial phenotypes and build stratified matured retina. Lower row: calcium-binding protein recoverin expression in regenerating retina cells in newt that reflects the stages of ONL formation [137]. DRPECs—dedifferentiating RPE-derived cells.

**Figure 6 biomedicines-10-01458-f006:**
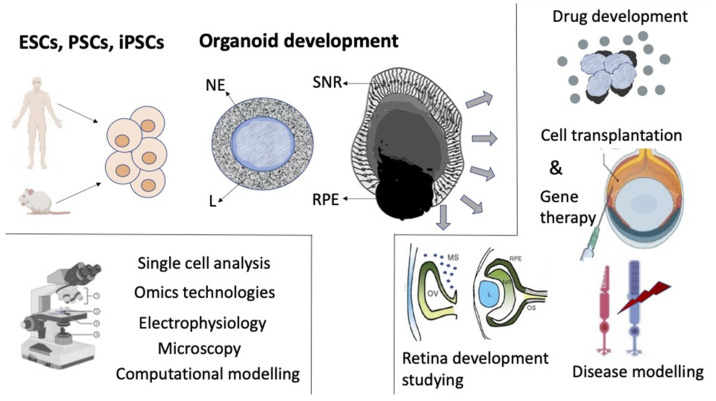
Three-dimensional retinal organoids obtained from pluripotent stem cells mimicking in vivo development. ESCs—embryonic stem cells; PSCs—pluripotent stem cells; iPSCs—induced pluripotent stem cells; NE—neuroepithelium; L—laminin; RPE—retinal pigment epithelium. SNR—stratified neural retina. Left box at the bottom: the methods of retinal organoids’ research. Right box: the directions of retinal organoids’ application.

## Data Availability

Not applicable.

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
