# Peer review of "Self-Organization of the Retina during Eye Development, Retinal Regeneration In Vivo, and in Retinal 3D Organoids In Vitro"

_biomedicines, 2022, doi:10.3390/biomedicines10061458_

Round 1
Reviewer 1 Report
This review recapitulated a massive number of literatures investigating the self-organization during eye development and retinal regeneration in vivo and in vitro. It mainly summarized the existing theories of self-organization in eye, which can be categorized into intrinsic factors, varieties of functional TFs and signaling pathways in a spatiotemporal order, and extrinsic factors, ECM, IOP, and RPE on how optic vesicle, optic cup, and retinal stratification are formed. I find it intriguing and profound that, in eye development, amphibian share a lot of same TFs and signaling pathways with those detected in vertebrates, which perfectly corroborates some conserved mechanisms in eye development among species and inspires further studies to uncover more potential molecules.
The authors did a comprehensive work, but several concerns were noticed when I read the manuscript.
Line 64-71: Based on the title of this study, high-quality articles specifically related to eye development and retinal regeneration (or retinal organoids) should be included. The following articles are recommended (but not limited with these):
Self-organizing optic-cup morphogenesis in three-dimensional culture. Nature. 2011 Apr 7;472(7341):51-6.
Self-formation of optic cups and storable stratified neural retina from human ESCs. Cell Stem Cell. 2012 Jun 14;10(6):771-785.
Stem Cell Res Ther. 2020 Aug 24;11(1):366.
Photoreceptor precursors derived from three-dimensional embryonic stem cell cultures integrate and mature within adult degenerate retina. Nat Biotechnol. 2013 Aug;31(8):741-7.
Sci China Life Sci. 2022 Apr 22. doi: 10.1007/s11427-021-2086-0.
Stemming retinal regeneration with pluripotent stem cells. Prog Retin Eye Res, 2019, 69: 38-56.
Proc Natl Acad Sci U S A. 2020 Dec 29;117(52):33628-33638.
Front Cell Dev Biol. 2020 Mar 6;8:128.
Line: 103-109: Lack of solid references to support these statements.
Figure 2B: It’s not indicative of RPE, NR, and optic stalk domains within the optic vesicle.
Line 134-137: Figure 3 cited here, which merely represents the timing of eye development, has nothing to do with TFs and other signaling molecules involved in differentiation. Figure3: The short black arrows are redundant when the long black arrow in the middle has indicated the time sequence of differentiation. Arrows in different colors, overlapping with each other, can be plausible but make it confusing to understand the so-called early and late phases. Which period in vertebrate embryo development can serve as the turning point between early and late phrases? Because there are no time windows of retinal cell differentiation provided in this review. To pinpoint the differentiation periods of each cell types can help with telling the time course.
Line 164: After the elimination of progenitors of ANY cell type in embryonic retina, what kind of cells were left to finally laminate? Line 193: More specific statements, with the key findings of your interest, could better be presented instead of just a simple mention.
Line 236-240: What are the detrimental outcomes of the genetically destroyed basal membrane of the neural crest? Like how does it affect the eye development in the mutant zebrafish? The answers are vital for proving the effect of neural crest ECM on a developing eye.
7. Line 287-314: The complex network of TFs regulating the early-stage eye development can be better illustrated with a figure, and it can also include how RPCs are determined, under other TFs or signaling pathways (Line 348-358), to differentiate into diverse retinal cell subtypes.
8. Please reorganize 3.2: The latter paragraphs are less relevant with the heading, but more dedicated to the techniques for analyzing genetic, epigenetic, or chromatin-level mechanisms in retinal development.
9. How do authors distinguish extrinsic factors in 3.2 and morphogenetic factors in 3.3? Morphogenetic factors, such as mechanical force from RPE, ECM stiffness, IOP and so on, are more like influences from external environment, which practically annotate extrinsic factors remotely addressed in 3.2.
1 Line 568: Reference numbers need to be scrutinized and corrected from here on. Recheck all the references because some don’t correspond with the context where they are cited. Meanwhile, make sure all references are cited in a consistent format.
1 Line 643: Better corrected as “retinal self-organizationin vitro”
1 Line 774-776: Cite organoid-related articles for how this term was coined, eg, Lancaster MA, Knoblich JA. Organogenesis in a dish: modeling development and disease using organoid technologies. Science. 2014 Jul 18;345(6194):1247125.
There are some grammatical errors, please reassess the article from start to finish. Also, for English language and style, there’s room for improvement.Some sentences are confusing, eg,line 35-39: Shouldn’t the studies of molecular genetics and epigenetic regulation be a part of the existing theories of biological self-organization? It’s not an either-or case. Examples like this are quite a few. So, the authors are suggested to specify their standpoints correctly through the whole article.
Author Response
Reviewer 1
I would like to express my deep gratitude to the reviewer 1 for the work on the article and the comments and suggestions made. Changes have been made and marked in the text in full compliance with the comments of the reviewer. All the changes and clarifications I made
are indicated in the paper. My responses to specific comments are below (italic)
Specific comments
Line 64-71: Based on the title of this study, high-quality articles specifically related to eye development and retinal regeneration (or retinal organoids) should be included. The following articles are recommended (but not limited with these):
Thank you very much for the references you recommended. Some of them have been already quoted in initial version of the review, others I included now. All of them are marked with numbers.
1.Self-organizing optic-cup morphogenesis in three-dimensional culture. Nature. 2011 Apr 7;472(7341):51-6. – this work has already been cited in the review (106)
- Self-formation of optic cups and storable stratified neural retina from human ESCs. Cell Stem Cell. 2012 Jun 14;10(6):771-785. – this work has already been cited in the review (107)
- Lancaster, M.L.; Knoblich, J.A. 2014. Organogenesis in a dish: modeling development and disease using organoid technologies. Science 2014, 345(6194), 1247125. doi: 10.1126/science.1247125. – this work included (195)
- Photoreceptor precursors derived from three-dimensional embryonic stem cell cultures integrate and mature within adult degenerate retina. Nat Biotechnol. 2013 Aug;31(8):741-7. – a later (2017) work by the same authors is cited (link 235).
- Sci China Life Sci. 2022 Apr 22. doi: 10.1007/s11427-021-2086-0 – This work is very special, dedicated to the development of methods for obtaining microglia from hPSCs. For this reason, I left it out of the review.
- Proc Natl Acad Sci U S A. 2020 Dec 29;117(52):33628-33638. Reference included (248)
- Front Cell Dev Biol. 2020, 8, 128. doi: 10.3389/fcell.2020.00128.Reference included (249)
- Retin Eye Res. 2019, 69:38-56. Reference included (252)
- Stem Cell Res Ther. 2020 Aug 24;11(1):366. Reference included (252)
Line: 103-109: Lack of solid references to support these statements. – references added
Figure 2B: It’s not indicative of RPE, NR, and optic stalk domains within the optic vesicle.
Changes have been made. NR, RPE, and optic stalk (OS) domains were indicated.
Line 134-137: Figure 3 cited here, which merely represents the timing of eye development, has nothing to do with TFs and other signaling molecules involved in differentiation.
Figure 3 was corrected. An Information about key transcription factors responsible for retinal cell type specification was added at the bottom of the figure. Moreover, I’ve prepared and added figure 4, where groups of TFs whose expression is associated with the main stages of the eye development are represented.
Figure3: The short black arrows are redundant when the long black arrow in the middle has indicated the time sequence of differentiation.
The thickness of the short black arrows has been reduced. However, they are left to show the conditional course of events.
Arrows in different colors, overlapping with each other, can be plausible but make it confusing to understand the so-called early and late phases.
It is usually manifested as the generation of ganglion cells, cone photoreceptors, and horizontal and amacrine cells of the retina in the early phase, overlapping with the late-phase generation of rod photoreceptors, bipolar cells, and Müller glia [17]. In accordance with this observation, I built the figure and drew a perpendicular line to separate (very conditionally) early and late phases. A dashed line was used for the demonstration.
Which period in vertebrate embryo development can serve as the turning point between early and late phrases? Because there are no time windows of retinal cell differentiation provided in this review. To pinpoint the differentiation periods of each cell types can help with telling the time course.
Early and late phases of retinal cell-types’ specification in mice correspond to the late embryonic and early postnatal periods, respectively. However, I have not found a strict binding of the maturation time windows of a particular cellular phenotype to a certain period of development. This is due to the fact that the cell-type specification periods overlap each other and quite differ in animals of different species. Usually, authors postulate that it occurs ‘relatively late’ in development [16-21].
Line 164: After the elimination of progenitors of ANY cell type in embryonic retina, what kind of cells were left to finally laminate?
To clarify, several sentences have been written and added to the text
Line 193: More specific statements, with the key findings of your interest, could better be presented instead of just a simple mention.
Two sentences are added just to emphasize the fact that programmed cell death is an integral component of self-organization. Detailed explanation of how this mechanism works is the subject of a separate cited review [40] (Vecino E.; Acera A. Development and programed cell death in the mammalian eye. Int. J. Dev. Biol. 2015, 59, 63-71).
Line 236-240: What are the detrimental outcomes of the genetically destroyed basal membrane of the neural crest? Like how does it affect the eye development in the mutant zebrafish? The answers are vital for proving the effect of neural crest ECM on a developing eye.
Two sentences added to explain the consequences of that mutation.
- Line 287-314: The complex network of TFs regulating the early-stage eye development can be better illustrated with a figure, and it can also include how RPCs are determined, under other TFs or signaling pathways (Line 348-358), to differentiate into diverse retinal cell subtypes.
In the last 30 years, a large number of TFs, combinations of genes, and also cofactors responsible for the eye domains (RPE, NR, OS etc.) and NR cell types’ specification were identified. It is big success of the study, but we are still far away from the complete picture of TFs regulating network at early stages of eye development as well as in time of retinal cell-types’ specification. Despite, the key ‘set’ of TFs participating in early eye development (figure 4) and retinal cell type specification (figure 3) were represented in the corrected version of the text.
- Please reorganize 3.2: The latter paragraphs are less relevant with the heading, but more dedicated to the techniques for analyzing genetic, epigenetic, or chromatin-level mechanisms in retinal development.
I did try to make section 3.2 more organized and divided it into two: “extrinsic factors and intrinsic factors” (3.2 and 3.3). I had in mind that the first ones are related to the microenvironment – factors produced by surrounding tissues, signaling pathways. The second refers to genome expression and epigenetic modifications. However, it is obvious that both categories of regulatory factors work together, in cohort, as a part of dynamic regulatory networks.
- How do authors distinguish extrinsic factors in 3.2 and morphogenetic factors in 3.3? Morphogenetic factors, such as mechanical force from RPE, ECM stiffness, IOP and so on, are more like influences from external environment, which practically annotate extrinsic factors remotely addressed in 3.2.
I completely agree. The selection of a group of "morphogenetic factors" is very conditional and is given by me in an attempt to stress out so called " mechanics" factors. However, at the same time I realize clearly that their effect is carried out through mechano-sensing mechanisms, behind which the work of signaling pathways inducing changes in gene expression and epigenetic landscape are standing.
1 Line 568: Reference numbers need to be scrutinized and corrected from here on. Recheck all the references because some don’t correspond with the context where they are cited. Meanwhile, make sure all references are cited in a consistent format.
All references and their citation in the text have been rechecked.
1 Line 643: Better corrected as “retinal self-organizationin vitro”
This subheading was corrected
1 Line 774-776: Cite organoid-related articles for how this term was coined, eg, Lancaster MA, Knoblich JA. Organogenesis in a dish: modeling development and disease using organoid technologies. Science. 2014 Jul 18;345(6194):1247125.
Reference included
There are some grammatical errors, please reassess the article from start to finish. Also, for English language and style, there’s room for improvement. Some sentences are confusing, eg,line 35-39: Shouldn’t the studies of molecular genetics and epigenetic regulation be a part of the existing theories of biological self-organization? It’s not an either-or case. Examples like this are quite a few. So, the authors are suggested to specify their standpoints correctly through the whole article.
The statement (line 35-39) is reworded
Grammatical errors were corrected, as well as style. Some disadvantages of the latter were noted by reviewer 2 and I tried to eliminate them.
Many thanks once more, E.N.Grigoryan

Reviewer 2 Report
Comments have been included in the pdf attached.

Author Response
Reviewer 2.
First and foremost, I express my deep gratitude to the reviewer for the work on the article and the comments made. Changes and corrections have been made and marked in the text in compliance with the comments. The changes made are indicated in the right margin of the paper. Responses to specific comments are below in italic.
Specific comments
Pg. 3, line 100, Section 3.1
- This section could be shortened and more targeted as majority of this concepts have been covered in detail in other reviews [https://doi.org/10.1016/j.conb.2014.02.014].
The section 3.1. provides basic/necessary information. I did try to present it shorter but I could not. Retinal self-organization in vivo is a complex and multifaceted process. My task of the review was, on the one hand, to explain this complexity, on the other, to make the explanation as short as it possible. The somewhat expanded paragraphs relate only to recent studies, in particular, on epigenetic regulatory mechanisms. Without them, the understanding of the versatility and complexity of the processes regulating self-organization at the modern level would be incomplete. In order to make the section more transparent, I introduced clarifications in Figure 3 and added Figure 4.
The paper Boije et al. (2014) as one of very important was cited in the discussion of two models of the specification of retinal cell types in early development.
Section 3.2. is divided into 3.2 (external factors) and 3.3 – (internal factors) as it was recommended.
Pg. 5, line 151, Figure 3
- The top arrows do not seem to be aligned to the cell types at the bottom. Consider re-aligning for better interpretation.
Figure 3 has been partially redone. The upper arrows are intentionally not aligned strictly according to the cell types. The maturation periods of one or another retinal cell-type overlap each other, and therefore, are not rigidly assigned to each cell type. Therefore, they are given in such a way in order to indicate the approximate period of сell-type specification. The binding of the specification time to the cell-type is given by the color. Inverted arrows at the bottom would change the structure and design of the figure.
Pg. 5, line 154, Figure 3 legend
- RPCs defined as retinal precursor cells. Consider standardizing RPCs to either retinal progenitor or precursor cells.
Fixed: ‘progenitor’ instead of ‘precursor’. Cells committed to produce specific retinal cell-types are indicated as ‘precursors’.
Pg. 6, line 224, Section 3.2
- This sub-section could be split into two different sub-sections, one focussed on intrinsic factors and other focussing on extrinsic factors.
Extrinsic and intrinsic factors are represented in separate subsections now.
Pg. 8, line 295, “Schematically, it can be represented as follows”
- Is a schematic missing?
“Schematically” is replaced by “in short”
Pg. 17, line 764 – 765, “The development of this area...”
- To be re-worded as sentence structure is unclear. The sentence is reworded.
Pg. 17, line 778 – 779, “These studies showed that not only complex...”
- Run-on sentence. The sentence is simplified and shortened.
Pg. 17, line 788 – 789
- Expand on SAG abbreviation. Abbreviation SAG is expanded
Pg. 18, line 805 – 807, “develops already at the first stages...”
- Sentence structure of the above.
I didn't quite understand what needs to be corrected here. Maybe this is a “roll call” with what is above, but it seems justified (?).
Pg. 18, line 807 – 808, “From a mechanical point of view...”
- Refers to zebrafish-derived retinal organoids. This makes the entire paragraph and explanation prior disjointed, as only mammalian retinal organoid models (mice and human) were being referenced.
Reference is changed.
Pg. 18, line 814 – 815, “The major issue of ensuring retinal histogenesis...for a rather long time”
- Sentence structure. Casual language.
Sentence is reworded
Pg. 18, line 816 – 818
- Ref 25 makes no mention of bioreactors, or any sort of retinal organoid cultivation. The other references refer to general reviews of various types of organoids. Thus, this entire sentence might not be needed.
Ref 25 is removed. It was entered by mistake. I left the sentence (lines 832-835) as it seems useful to readers, as a brief reference on cultivating in vitro specifically with bioreactors.
Pg. 18, line 819 – 820
- Very general statement. Can be excluded or shortened.
(please see below)
Pg. 18, line 820 – 825, “Thus, the major issue is the reconstruction...”
- This conclusion seems to be out of place, without any prior expansion and explanation. There was also no specific reference to these points regarding retinal organoids, as the two references given were reviews, instead of the original research study.
Fragment of the text (lines 838-841) is rewritten and shortened.
Pg. 18. line 836 – 838, “Earlier, the case of 2D cultures produced...”
- Casual language.
Sentence is reworded (language)
Is the effect of IGF-1 also induced by inhibition, as with BMP and Wnt? Or is it something completely different?
In the text of the paper (Lamba et al., 2006) the next is written: “…we treated embryoid bodies with a combination of noggin (a potent endogenous inhibitor of the BMP pathway) and Dickkopf-1 (dkk1; a secreted antagonist of the Wnt/β-catenin signaling pathway (14, 21)) and IGF-1.
As it follows from that, the positive effect of IGF-1 was independent after supplementation of IGF1 to hPSC-derived Noggin/Dkk1 in vitro cultures. I didn’t see there any reasonings about the promotion of IGF-1 positive effect by suppression of BMP and Wnt/β-catenin signalings.
It is also known that the inhibition of IGF‐1 receptor signaling significantly reduces the formation of optic vesicle and optic cups, while exogenous IGF‐1 treatment enhances the formation of correctly laminated retinal tissue composed of multiple retinal phenotypes that is reminiscent of the developing vertebrate retina (Mellough et al., 2015).
Pg. 18 – 19, line 845 – 847
- Specific studies should be referenced instead of a review, alongside an expansion on what these redundancies and plasticity in pathways are.
A detailed analysis of the differences in the work of signaling pathways in the formation of organoids and in the development of the retina has already been carried out by O'Hara and Gonzalez-Cordero, 2020. For this reason, I made corrections to the wording without detailing.
Pg. 19 – 20, line 899 – 901
- Unclear sentence. The sentence is rewritten
Pg. 20, line 919 – 920
- Improper citation location of Ref 243. Reference is replaced
Pg. 20, line 924 – 932
- Improper grammar, vocabulary, run-on sentence
“On the one hand, it is a convenience for studying slowly maturing cell types, the work of molecular regulators of the process, as well as the slowly forming phototransduction process. On the other hand, the duration of cultivation requires long-term availability of conditions required not only to maintain the viability of the tissue, but also to provide its development on a spatial-temporal way. Observations have been published that the process of maturation of retinal organoids occurs faster under hypoxia conditions, which are known to exist during the eye development in vivo [245,246]. It is also known that, when exposed to the hypoxia conditions, PSCs, which are a source of RPCs in the case of the formation of organoids, are less stressed, divide faster, have less cleaved Cas-3, and exhibit fewer chromosomal abnormalities [245].”
This paragraph is reworded
Pg. 20 – 21, line 951 – 974
- The multiple paragraphs starting from line 951, and ending at line 974, should be rewritten, and used as an introductory paragraph for section 5.2. The paragraph starting from line 969, also feels out of place as a conclusion to the whole section on retinal organoids.
These paragraphs are devoted to the broad prospects of the use of retinal organoids. Various possibilities of using retinal organoids are the result and success of long-time study of retinal self-organization in vivo and in vitro. The paragraphs are presented in the most concise form and they still seem for me more appropriate at the end of section 5.2. Discussion of the use of organoids at the beginning of the section, before explaining the methods of their obtaining and results of retinal organoid study, as I think, would be more complicated for readers.
Many thanks again, E.N.Grigoryan

Round 2
Reviewer 1 Report
The authors addressed most of my concerns. However a point of problem in the rebuttle letter remains.
The authors exclued the reference (PMID: 35451725) in the manuscript with the reason I can NOT agree with. This study describes a new and more realistic retinal organoid with integrated microglia. Actually, neuroectoderm-derived retinal organoids do NOT have any microglia which is from york sac. In this reference, newly differentiated microglia were self-assemblied into conventional retinal organoids, empowing more emulational retina with the important neuroimmune cells.
Author Response
I express my great gratitude for the review once again. The recommended work (Gao et al., 2022) is included now in the paper: short description of this study is in lines 941-947; reference number 244 in the text and reference list.
Errors and typos have been corrected once again throughout the text.
Reviewer 2 Report
The author has made significant changes based on the comments and recommendations that answers my points satisfactorily. This review will be an important contribution to the field of retina developmental biology and how its understanding can be used to the advantage for translational applications. It was a pleasure reviewing it.
Author Response
Thank you again for your attention to the manuscript and for detailed review.